# Mechanically Sustainable Starch-Based Flame-Retardant Coatings on Polyurethane Foams

**DOI:** 10.3390/polym13081286

**Published:** 2021-04-15

**Authors:** Kyung-Who Choi, Jun-Woo Kim, Tae-Soon Kwon, Seok-Won Kang, Jung-Il Song, Yong-Tae Park

**Affiliations:** 1School of Aerospace and Mechanical Engineering, Korea Aerospace University, 76 Hanggongdaehak-ro, Deogyang-gu, Goyang-si 10540, Gyeonggi-do, Korea; kwchoi@kau.ac.kr; 2Department of Mechanical Engineering, Myongji University, 116 Myongji-ro, Cheoin-gu, Yongin 17058, Gyeonggi-do, Korea; rlawnsdn418@gmail.com; 3Korea Railroad Research Institute, 176 Cheoldo bangmulgwan-ro, Uiwang-si 16105, Gyeonggi-do, Korea; klez@krri.re.kr; 4Department of Automotive Engineering, Yeungnam University, 280 Daehak-ro, Gyeongsan 38541, Gyeongsangbuk-do, Korea; swkang@yu.ac.kr; 5Department of Mechanical Engineering, Changwon National University, 20 Changwondaehak-ro, Uichang-gu, Changwon 51140, Gyeongsangnam-do, Korea; jisong@changwon.ac.kr

**Keywords:** layer-by-layer assembly, flame retardant, cationic starch, montmorillonite clay, eco-friendly, durable, polyurethane foam

## Abstract

The use of halogen-based materials has been regulated since toxic substances are released during combustion. In this study, polyurethane foam was coated with cationic starch (CS) and montmorillonite (MMT) nano-clay using a spray-assisted layer-by-layer (LbL) assembly to develop an eco-friendly, high-performance flame-retardant coating agent. The thickness of the CS/MMT coating layer was confirmed to have increased uniformly as the layers were stacked. Likewise, a cone calorimetry test confirmed that the heat release rate and total heat release of the coated foam decreased by about 1/2, and a flame test showed improved fire retardancy based on the analysis of combustion speed, flame size, and residues of the LbL-coated foam. More importantly, an additional cone calorimeter test was performed after conducting more than 1000 compressions to assess the durability of the flame-retardant coating layer when applied in real life, confirming the durability of the LbL coating by the lasting flame retardancy.

## 1. Introduction

Many studies on environment-friendly flame retardants have been conducted recently due to issues related to human hazards and environmental pollution. Although halogen and bromide flame retardants were mainly used in the past, their use has been restricted by an environmental regulation because they generate gas that corrodes metals and which is hazardous to the human body during combustion [1,2,3,4]. As such, studies to find alternative flame retardants have been conducted, along with studies on improving flame resistance using properties such as low heat discharge [5,6,7,8]. This study analyzed the flame-retardant property of layer-by-layer (LbL) assembly on polyurethane (PU) foam using cationic starch (CS)–montmorillonite (MMT) clay. The LbL assembly creates a stable and thermally low-conductive char layer on the surface of the PU foam during pyrolysis.

Developed in the early 1990s, the LbL assembly is a method of using polymer electrolytes [9,10,11]. By using positively or negatively charged polymer electrolyte, it generates thin, multi-layered films stacked on the surface of various materials through electrostatic interaction, hydrogen bonding, hydrophobic interaction, or covalent bonds in aqueous solution. It can be applied in a wide range of areas thanks to its easy and simple manufacturing and its advantage of controllable surface treatment, such as the thickness and roughness of thin films, depending on the materials used [12,13]. Currently, LbL assembled thin films can be applied to various substrates ranging from glass, silicon wafers, and nano-scale to macromolecular polymer thin films. There are many studies on electrical conductivity [14,15,16], antibacterial [17,18], gas barrier [19,20,21], detection [22,23,24], drug delivery [25,26,27], and flame retardants [1,4,28,29,30,31].

Recently, there has been increasing interest in flame retardant coatings with various techniques including LbL assembly. Zhang et al. synthesized eco-friendly biomass coating on cotton fabrics with tannin (TA), tartar emetic (TE), and Fe^2+^ by a similar process to dye-fixing. The limiting oxygen index (LOI) value was almost not changed after 100 laundering or friction cycles due to strong hydrogen bonds and electrostatic interactions among TA, TE, and Fe^2+^ [32]. Fang et al. applied intumescent flame retardant coating of chitosan (CH)/phytic acid (PA) by LbL process on polyester/cotton blend fabric. Due to prompt char formation induced by CH/PA coating, the LOI value of 20 bilayers coated sample reached 29.2% [33]. Meanwhile, a rapid dip-coating process was used to prepare a flame-retardant coating for polyethylene terephthalate (PET) fabrics by Tao et al. [34]. The total heat release rate of coated PET was reduced to 4.8 MJ/m^2^ whereas that of uncoated PET was 7.1 MJ/m^2^. This was caused by worse thermal degradability and char formability of coatings. Yu et al. constructed graphene oxide nanoribbon/MMT/polyethylene glycol coatings on PU foam. Hybrid networks such as nacres showed enhanced flame resistance and structural stability in flames due to the formation of interconnected compact charcoal after combustion [35]. However, until now, few studies have quantitatively investigated whether the flame retardant effect is maintained after repeated compression tests on foams with flame retardant coatings.

PU foam (PUF) is one of the most widely used interior materials due to its excellent resilience and mechanical property. Nonetheless, it has high fire vulnerability due to its high ignitability and fast flame propagation speed. When the PUF is ignited, it creates flame droplets that easily spread fire with nearby combustibles and produces a lot of harmful smoke. In order to improve the flame-retardant performance, flame retardants such as phosphorous, nitrogen, or halogen fillers can be incorporated into the PU foam [36,37,38].

In this research, PU foam was coated with CS and MMT clay to functionalize the eco-friendly flame-retardant characteristics. CS is a bio-based material that shows a positive charge when dissolved in water, whereas MMT clay is an eco-friendly flame retardant that shows a negative charge when dissolved in water. PU foam was coated with these two materials in five bilayers using the electrostatic properties of the LbL assembly of positively and negatively charged materials in turn. While dip coating is widely used for LbL assembly, spray coating was used in this study for thin-film coating. Spray coating is more effective for industrial applications, such as coating of large materials and machine automation, as it is superior to dip coating in many ways. Also, for the first time, we report the lasting flame retardancy of LbL-assembled multilayers on PU foam after 1000 cycles of compression. The degree of flame propagation, mass, and heat release after combustion were checked using the open flame test, thermogravimetric analysis (TGA), and cone-calorimetry

## 2. Experimental

### 2.1. Layer-by-Layer Assembly

MMT clay was purchased from BYK in Korea. The density of MMT is known to be about 2.86 g/cm^3^, and it shows a negative charge when dissolved in water. CS was purchased from Samyang Corporation (Sun-casta, Korea). It shows a positive charge when dissolved in water. The CS was dissolved in deionized (DI) water to a concentration of 0.1 wt % at a temperature of 90 °C, and the MMT clay was dissolved in DI water to a concentration of 0.1 wt %. A hot plate and a stirrer were used to dissolve the two materials in DI water for a day to disperse them sufficiently. Once fully dissolved, CS was sprayed on the PU foam while maintaining the temperature, and then rinsed with DI water to remove the excess solution. Flexible PU foam (tradename: RJBB, 0.027 g/cm^3^) was purchased from Kyunggi Sponge (Gyeonggi-do, Korea). While coating on both sides of PU foam, an air pressure of 3.0 bars was continuously supplied to the air-assisted dual fluid sprayers, which corresponds to a spray liquid volume flow rate of about 3.0 L/h. Compared with a conical nozzle that generates a centralized spray, a fan-shape nozzle was used to provide a uniform spray to achieve a uniform coating surface. The distance from the spray nozzle to the substrate was fixed at 10 cm. Next, MMT clay was sprayed on it and rinsed with DI water. Each positive and negative pair deposited is composed of an LbL-assisted thin film, referred to as a bilayer (BL), and this process was repeated to stack 5 BLs (Figure 1). After the completion of the 5 BLs, the coated PU foam was maintained at a temperature of 80 °C, and then placed in a vacuum oven to dry for about a day.

### 2.2. Measurements and Characterization

An ultraviolet–visible (UV–Vis) spectrometer (DH-2000-BAL, Oceans Optics) was used to measure the absorbance of PET film coated with 5, 10, and 15 BLs each of CS/MMT. The mass of the coated layers was measured by a quartz crystal microbalance (QCM, USB-2000, Ocean Optics) and 5 MHz gold-electrode quartz crystals. The QCM crystal surfaces were cleaned by oxygen plasma prior to use. To qualitatively evaluate the flame retardant performance of the coated foam (100 × 100 × 30 mm^3^), the flame was applied directly to the center of sample sidewall using a butane gas hand torch. The sample was placed horizontally on a metal grid inside the fume hood and the side end was 10 mm next to the torch tube. A 20 mm-long blue flame is applied to the center of the sidewall for 10 s and then removed. When the PU foam melted and fell, the PU drops fell on a layer of dry surgical cotton placed 300 mm below the grid. The thermal stability of the sample was tested using TGA (DSC 823e, Mettler Toledo). 100 mg samples were prepared and measured by increasing the temperature from 30 °C to 800 °C at a rate of 10 °C/min under air and nitrogen atmosphere. The morphologies of the neat, coated foams, and char residue after the flame test were examined using a field emission scanning electron microscope (FE-SEM, SU-70, Hitachi) operating at 10.0 kV. Each sample was coated with 4 nm-thick Au before imaging. Cone calorimetry was performed based on ISO-5660-1. The heat release rate (HRR), total heat release (THR) and maximum average rate of heat evolved (MARHE) were measured by transferring 25 kW/m^2^ radiation heat to a 100 × 100 × 30 mm^3^ sample. Compression tests were performed on the 5 BL coated foams to analyze the durability of the spray-assisted LbL coatings on PU foams. To test the flame retardancy after 1000 cycles of compression, the specimen was fabricated in the identical size as the cone calorimeter samples. The compression was performed according to ASTM D 3574 C. By using a universal tensile test machine (AGS-X, Shimadzu Scientific Korea), the specimen was recovered after compression by 50% deformation over the entire top surface of the specimen, and this was designated as 1 cycle.

## 3. Results and Discussion

### 3.1. Spray-Assisted LbL Thin Film Growth

The monochromatic light absorbance of the material is proportional to the light transmission depth of the sample layer. As such, it is possible to check each layer of coating on the samples using UV–Vis. The growth of the thin film was verified by using the sample whose PET film was coated with 5, 10, and 15 BLs of CS/MMT. Figure 2a shows the absorbance at 550 nm wavelengths in the control, 5, 10, and 15 BL samples, and the inset graph illustrates the absorbance of 400–800 nm of the number of BLs. A neat PET substrate was considered the basis for the UV–Vis absorbance spectrum. The UV–Vis results were averaged based on three measurement results of each BL sample. The fact that the degree of light absorbance increases as the BL is deposited shows that the thickness of the coating layer increases. Moreover, from the linearity of the light absorbance, it can be assumed that the film growth on the substrate is well-controlled as the layers are increased. Figure 2b shows the mass increase of [CS/MMT]*_n_* layers with a QCM to indicate the coating growth of the film. The graph is a plot of measurements of the layers of CS and MMT and is similar to the results obtained by UV–Vis. The amounts of CS and MMT adsorbed onto the quartz crystal in each deposition cycle were estimated to be similar. Here, the slope of 5–10 BLs differs from that of 10–15 BLs. The change of slope in the QCM or UV–Vis plots is deemed likely to be due to the mutual dispersion of weak polymer electrolytes as reported in previous studies on LbL films [21]. In summary, UV–vis and a quartz crystal microbalance were used to measure the linear growth of the LbL film as a function of the deposited bilayer.

### 3.2. Thermal Stability

As shown in Figure 3, the thermal stabilities of the control and coated samples were measured with TGA under nitrogen and air atmosphere, while the representative thermal parameters are summarized in Table 1. In an inert atmosphere, both control and coated samples showed two major thermal degradation stages with the initial degradation at 290–310 °C (T_max1_) and the second degradation at 370–380 °C (T_max2_). These correspond to the decomposition of urethane and urea and the pyrolysis of the polyether backbone, respectively. At 800 °C, 0.7 wt % and 1.2 wt % of residue of the control sample remained in air and nitrogen, respectively, and 2.0 wt % and 5.6 wt % of residue of the 5 BL-coated sample remained in air and nitrogen, respectively. The change in mass of the control and 5 BL samples differs at 400 °C or higher, with higher mass of the 5 BL sample remaining. These results indicate that the char generated when the outermost CS/MMT composite layer of the PU foam is thermally decomposed, protecting the flammable substances by blocking out oxygen. Moreover, the inset graph (derivate weight loss, DWL) of Figure 3 shows the thermal change behavior of the PU foam. Under an air atmosphere, the control sample showed two peaks at around 300 °C, whereas the 5 BL sample showed a large peak before a rapid decrease. This indicates that the 5 BL sample thermally stabilizes as the quickly burnt flame-retardant coating layers form a char layer and prevent combustion of the foam. Such a phenomenon is confirmed by the second peak under a nitrogen atmosphere. The 5 BL sample and control showed a similar peak at 300 °C under a nitrogen atmosphere unlike an air atmosphere, indicating that the bonds of PU are being destroyed. The graph also shows another decrease in mass at around 500 °C under an air atmosphere, namely natural thermal oxidation, while no peak at around 500 °C in a nitrogen atmosphere [40,41,42].

Cone-calorimeter tests were performed to measure the heat release rate for the quantitative analysis of flame retardance. In this test, the samples were exposed to radiation heat rather than direct flame. Figure 4 shows the HRR and THR of control and 5 BL sample as functions of time at a constant heat flux of 25 kW/m^2^. Figure 4a and Table 2 illustrate that 5 BL sample showed peak HRR value of 147.0 kW/m^2^ which was 77.3% of that of control sample and then rapidly dropping afterward. Such a rapid drop of HRR is due to the carbon layer created on the outermost surface of the foam during combustion, which blocks heat discharge and prevents internal combustion. Maximum average rate of heat emission (MARHE) of 5 BL sample was 97.2 kW/m^2^ while that of the control sample showed 138.5 kW/m^2^. Moreover, the THR of the 5 BL sample decreased by about 50% compared to the control.

In order to verify the resilience of the coating layers, HRR and THR were obtained (Figure 4b) after 1000 cycles of compression test performed by compressing the samples to 50% of their thickness and restoring them (ASTM D3574 C, Foam Force Deflection Testing). The peak heat release rate (pkHRR) of the compressed 5 BL sample increased to 195.3 kW/m^2^, which was 133% of the uncompressed 5 BL foam, and even higher than pkHRR of control sample. However, the MARHE of the compressed 5 BL sample increased by only 5.5% compared to the uncompressed 5 BL sample and the THR was even reduced to 17.8 MJ/m^2^ (89.4% of uncompressed 5 BL sample). When compared to the control sample, the flame retardance of compressed 5 BL coated foam was still superior to that of the control. This confirms that the samples of coated PU foam remained without being detached by external physical force, thereby maintaining their flame-retardant effect.

### 3.3. Flame Test

The control and 5 BL samples were subjected to the flame test to check their flame dispersion and anti-flammability characteristics. Flammability testing was performed by exposing the neat and coated PU foams directly to a butane torch in a horizontal setting. The effect of flame retardant coatings was investigated by reducing melt dripping, spreading flames, preserving shape and forming char. Figure 5a,b show the control sample 5 and 15 s after combustion, respectively, whereas 5c and d show the 5 BL sample 5 and 15 s after combustion, respectively. During flame exposure, a pristine PU foam rapidly ignited and then immediately melt dripped. Figure 5a,b show the flame propagation of the control as the foam melted at the same time as the ignition with high flame propagation speed and vigorous flame. Furthermore, the sample was completely burnt away, leaving no residue. This phenomenon is the typical burning behavior of PU foam and is one of major problems as it tends to transmit flame to the surrounding combustible materials. In contrast, as shown in Figure 5c,d, the 5 BL sample had a relatively small and slower propagation of flame, and its inside was well-preserved even after combustion. The only 5 BL of clay coating prevented the melt from dripping, and the flame propagated across the entire foam surface and then extinguished itself. Figure 5e,f show that the char layer was generated on the outer surface during combustion, and that the inner structure was not damaged. Figure 5g,h present the flame test results of the 5 BL-coated PU samples cut in half. Even though the coated foams were burned after cutting, the inside of the 5 BL-coated PU foam was well-preserved similar to the uncut sample. This implies that the spray-assisted LbL was successfully deposited inside the structure.

The surfaces of all the samples were observed with SEM to examine the microscopic porous structures of the control sample and the LbL coating morphology on them. Figure 6 shows the SEM images of the control and coated samples before and after the flame test. The control showed holes with softer and smoother surfaces inside the foam, whereas the sample coated with [CS/MMT]_5_ showed a rougher surface than the control, with CS and MMT coating the wall of the hole inside the foam. Figure 6e,f show the coated sample after the flame test. Inside the foam, the structure of the outer part was damaged by combustion, whereas the inner part hardly showed any change compared with the image before combustion; thus, confirming that the inner structure was well-preserved. For further analysis, the SEM images of the boundaries outside and inside of the sample after combustion were observed. Figure 7 shows that the outermost part of the sample was significantly damaged, whereas the internal foam structure was well-preserved. Thus, although the foam structure was damaged by combustion in the boundary area, its initial structure was maintained. These images confirm anew that combustion from the outside generates char layers and protects the inside structure.

## 4. Conclusions

This study proposed an effective way to increase anti-flammability of PU foam through the spray-assisted LbL assembly of an eco-friendly flame retardant. CS and MMT were used as eco-friendly retardants to form a combustion-protective layer using the LbL assembly method. The deposition of the layers was confirmed by UV–Vis, QCM, and SEM images. The optical characteristics and increase in mass of the film indicated the growth of the LbL assembly on the substrate.

The results of the TGA showed that 1.26 wt % and 4.4 wt % more mass of the 5 BL-coated sample remained under an air atmosphere and a nitrogen atmosphere, respectively, compared to the control when the samples were heated to 800 °C. This shows that a char layer was formed to protect the inside and block the progress of combustion. The cone calorimeter test showed that the HRR and THR of the 5 BL-coated sample were about 1/2 less than those of the control. Amazingly, the THR of the 5 BL-coated sample after 1000 cycles of compression test was about 46.5% compared to control, showing the excellent durability of the spray-assisted LbL coatings on the PU foam. The result of the flame test showed that the foam of the control melted without any residue, but that of the 5 BL sample showed a much smaller flame with slower propagation speed while maintaining its feature. The results of the TGA, cone calorimeter, and flame tests showed that a char layer was formed on the coated sample during combustion to preserve the inside structure and suppress the progress of combustion. In conclusion, it is believed that the LbL coating of the widely used PU foam would be effective in protecting crops with an eco-friendly flame retardant.

## Figures and Tables

**Figure 1 polymers-13-01286-f001:**
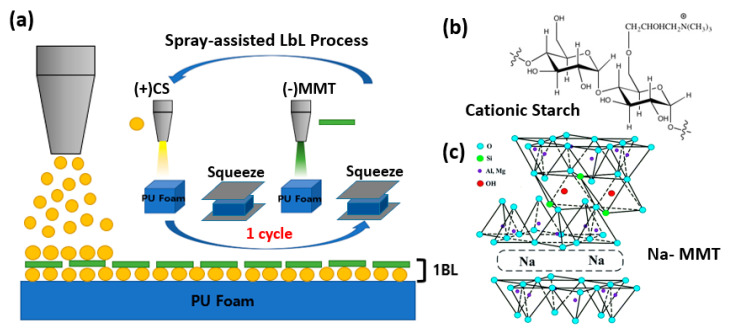
(**a**) Schematic illustration of the spray-assisted layer-by-layer assembly of cationic starch–montmorillonite [CS/MMT]; (**b**,**c**) chemical structures of cationic starch and sodium montmorillonite platelet shown here is reprinted with permission from Reference [39].

**Figure 2 polymers-13-01286-f002:**
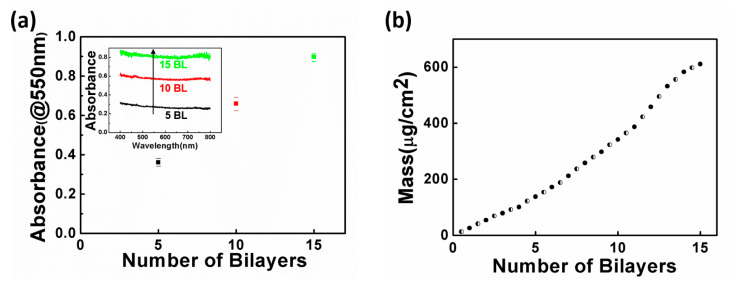
(**a**) The change in absorbance (measured at 550 nm) with an increase in the number of bilayers (BLs). The inset is ultraviolet–visible (UV–Vis) absorbance spectra of [CS/MMT]*_n_* (*n* = 5, 10, 15) thin films and (**b**) quartz crystal microbalance (QCM) mass of [CS/MMT]*_n_* as a function of the number of BLs, where CS mass deposition is denoted as half-filled points and MMT as filled points.

**Figure 3 polymers-13-01286-f003:**
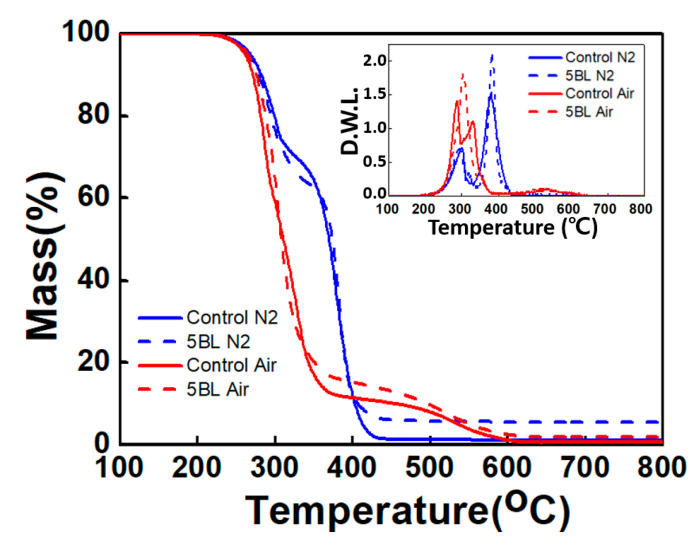
TGA mass loss of untreated and LbL-treated samples under nitrogen and air conditions, respectively. The inset is derivate weight loss plots.

**Figure 4 polymers-13-01286-f004:**
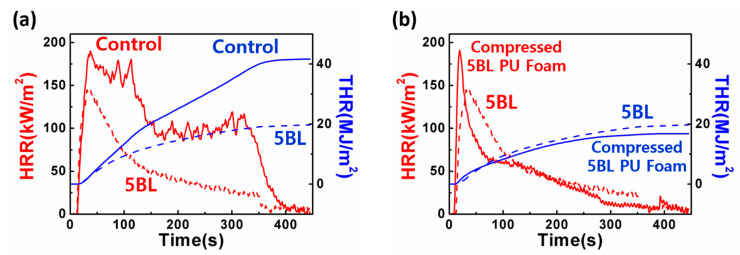
(**a**) Heat release rate (HRR) and total heat release (THR) plots of control and [CS/MMT]_5_-coated PU foams; (**b**) HRR and THR plots of [CS/MMT]_5_-coated PU foam before and after being compressed 1000 times.

**Figure 5 polymers-13-01286-f005:**
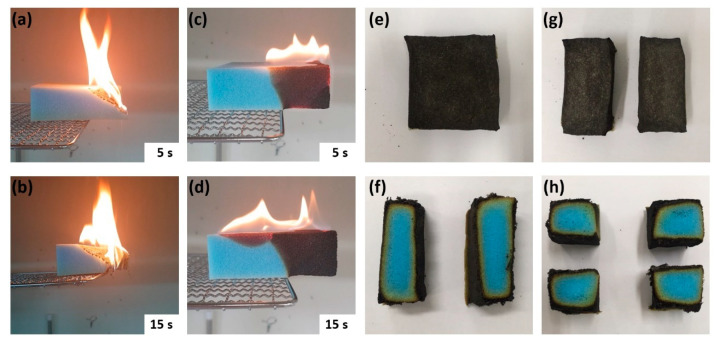
Snapshots showing flame tests of (**a**,**b**) the control and (**c**,**d**) layer-by-layer (LbL)-coated samples recorded at 5 and 15 s after ignition. (**e**,**f**) Images of outer and inner surfaces of the 5 BL-coated PU foam after the flame test. (**g**,**h**) Post flame test images of outer and inner surfaces of 5 BL-coated samples cut in half.

**Figure 6 polymers-13-01286-f006:**
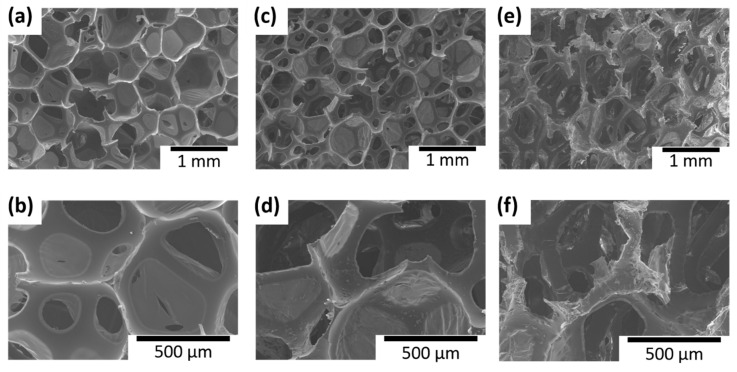
Scanning electron microscope (SEM) images of inner foam structures of (**a**,**b**) the control sample and (**c**–**f**) the 5 BL-coated sample (**c**,**d**) before and (**e**,**f**) after the flame test.

**Figure 7 polymers-13-01286-f007:**
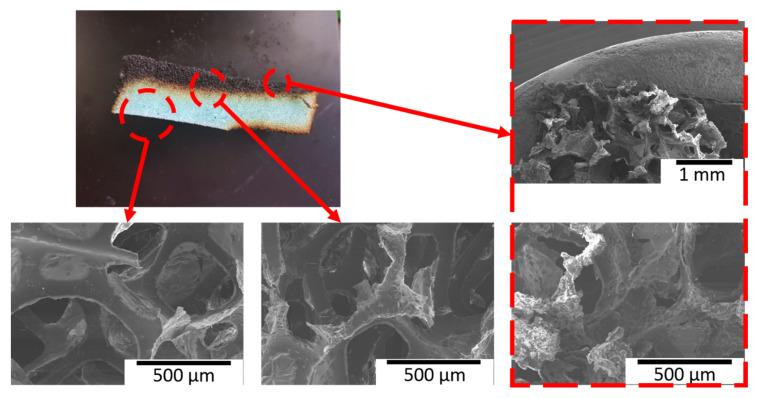
SEM images of the 5 BL-coated sample after the flame test.

**Table 1 polymers-13-01286-t001:** Thermogravimetric analysis (TGA) data of pristine and coated polyurethane (PU) foams in N_2_ and air.

Sample	T_−2%_ (°C)	T_−10%_ (°C)	T_max1_ (°C)	T_max2_ (°C)	Residue at 800 °C (%)
Under N_2_					
No coating	250	290	300	372	1.2
5 BL coating	250	288	298	380	5.6
Under air					
No coating	245	275	291	332	0.7
5 BL coating	248	282	302	-	2.0

**Table 2 polymers-13-01286-t002:** Cone calorimetry (ISO 5560-1) results of the control and 5 BL coated PU foams before and after 1000 compressions

Sample	phHRR (kW/m^2^)	MARHE (kW/m^2^)	THR (MJ/m^2^)
Control	190.2	138.5	42.1
5 BL-Before	147.0	97.2	19.9
5 BL-After	195.3	102.5	17.8

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
