# Peer review of "Mechanically Sustainable Starch-Based Flame-Retardant Coatings on Polyurethane Foams"

_polymers, 2021, doi:10.3390/polym13081286_

Round 1

Reviewer 1 Report

" A hot plate and a stirrer were used to dissolve the two materials in DI water for a day to disperse them sufficiently." - is it actually enough to disperse the MMT properly? Any results? Many works indicate the need of additional treatment e.g. by ultrasounds.

Why concentration of 0.1% was used? And why only one concentration was analyzed? Any previous works?

Why only 5 BL were analyzed further?

Any evidence that the growth of layers occurrs in the similar way on PET and PU foam?

Please provide also T-1% or T-2% from TGA analysis, which are ofter referred as onset of decomposition.

How 1000 compressions could reduce the THR of coated material?

Author Response

Reviewer: 1

Comments and Suggestions for Authors

We appreciate the reviewer’s important comments and suggestions. The authors thoroughly revised the manuscript and added relevant introductions, references, and discussions. The significant enhancement in anti-flammability of nanoclay-based coating was investigated.

  1. " A hot plate and a stirrer were used to dissolve the two materials in DI water for a day to disperse them sufficiently." - is it actually enough to disperse the MMT properly? Any results? Many works indicate the need of additional treatment e.g. by ultrasounds.

We appreciate the reviewer’s important comment. MMT can be exfoliated by simply adding it to DI water under magnetic stirring for 24 h. We used the sodium-montmorillonite and it was well dispersed in DI water without further treatment. The spray-based coating required a large amount of solution, so tip-ultrasonication would not be a good treatment method. Compared to bath-sonication, vigorous magnetic stirring showed better MMT dispersion in DI water. Additionally, some previous papers, including our group papers, have reported that Na-MMT was well dispersed in water without additional treatment.

  1. Why concentration of 0.1% was used? And why only one concentration was analyzed? Any previous works?

We appreciate the reviewer’s important comment. In a previous study, a 1.0 wt% MMT solution was used because MMT was coated on PET sheet or cotton fabric by dip coating. After dipping, the sample was rinsing with DI water several times and drying with the compressed air to remove the excess molecules/particles. In this study, however, PU foam has many pores inside and outside, making it difficult to remove the excess material. Because (1) we already knew the higher concentration solution was available to LbL coating, (2) PU is a porous substrate, and (3) the spray-based coating required a large amount of solution, we considered the efficient coating process. Thus, we used 0.1 wt% aqueous solutions for spray-assisted LbL coating in this study.

  1. Why only 5 BL were analyzed further?

We appreciate the reviewer’s important comment. In this study, we found 5 BL was sufficient to suppress the burning of PU foams, and more BL coatings also showed similar burning behavior through flame tests. Thus, we did not perform the further analysis such as TGA and cone calorimetry, etc.

  1. Any evidence that the growth of layers occurrs in the similar way on PET and PU foam?

We appreciate the reviewer’s important comment. The authors also know that since the quantitative thickness increase can be measured only on a very limited substrate (i.e., 2-dimensional flat hard substrates), it is difficult to accurately determine the thickness of the PU foam. Nevertheless, since it is possible to grasp the growth of the thin film due to the chemical attraction between the two counterparts, it has been used as an analysis method for the increase in thickness. Therefore, many previous studies of LbL coatings on 3D substrates have conducted experiments and analysis in this way.

  1. Please provide also T-1% or T-2% from TGA analysis, which are ofter referred as onset of decomposition.

We appreciate the reviewer’s important comment. The T-2% values were added in the manuscript. The T-2% values of each samples under N2 and air showed similar, so the onset of decomposition begins at the same temperature.

  1. How 1000 compressions could reduce the THR of coated material?

We appreciate the reviewer’s important comment. It is because after compression the clay coating did not decompose and the coating adhesion to PU substrate was well maintained. Also, the authors assumed that PU foam is porous, so the LbL thin layer was not compressed (i.e., no stress applied) during the compression test.

Reviewer 2 Report

Reviewers' comments:

Manuscript number: polymers-1155617

Title: Mechanically sustainable starch-based flame-retardant coatings on polyurethane foams.

Comments: 

The manuscript reported on Mechanically sustainable starch-based flame-retardant coatings on polyurethane foams. The manuscript needs a detailed editing.

- English corrections is required throughout the manuscript.

- Add more keywords.

- The introduction section should be improved; more related papers must be discussed and superiority, novelty, critical improvement in this study must be clarified.

- The figures are too pixelated to be useful. Please provide sharper pictures with better resolution.

- The experimental section should be detailed especially for the surface techniques. 

- Figure 1 – not clear make clear.

- 3.1. Spray-assisted LbL Thin Film Growth - must be improved.

- 3.3. Flame Test - must be improved.

- In part SEM: how the energy of the accelerator beam used?

- Several faults: are added or missing spaces between words: see PDF file.

- Conclusions, the author should add some qualitative data of the results.

- References: there are recent references in 2020 and 2021 treating the same subject, you can use.

So that I recommended this manuscript to major revision and for future process.

Author Response

Referee: 2

Comments to the Author

The manuscript reported on Mechanically sustainable starch-based flame-retardant coatings on polyurethane foams. The manuscript needs a detailed editing.

  1. English corrections is required throughout the manuscript.

We appreciate the reviewer’s important comment. As mentioned above, we have carefully revised this manuscript and it was given a proof of English by a professional company.

  1. Add more keywords.

We appreciate the reviewer’s important comment. We added more keywords, so there are total 7 keywords “Layer-by-layer assembly; Flame retardant; Cationic starch; Montmorillonite clay; Eco-friendly; Durable; Polyurethane foam.”

  1. The introduction section should be improved; more related papers must be discussed and superiority, novelty, critical improvement in this study must be clarified.

We appreciate the reviewer’s important comment. The authors added additional descriptions of several recent papers related to this study and showed the novelty of this study in Introduction. Also, the authors improved relevant points and references in the introduction section.

  1. The figures are too pixelated to be useful. Please provide sharper pictures with better resolution.

We appreciate the reviewer’s important comment. As a reviewer’s recommendation, we changed the figures clearer to easily show and find the actual values.

  1. The experimental section should be detailed especially for the surface techniques.

We appreciate the reviewer’s important comment. The authors modified some surface characterization methods such as UV-vis, QCM, and FE-SEM.

  1. Figure 1 – not clear make clear.

We appreciate the reviewer’s important comment. As a reviewer’s recommendation, we changed Figure 1 clearer.

  1. 3.1. Spray-assisted LbL Thin Film Growth - must be improved.

We appreciate the reviewer’s important comment. UV-vis and a quartz crystal microbalance were used to measure the linear growth of the films as a function of bilayers deposited, while scanning electron microscopy was used to visualize the nanostructure of the foams. The authors improved the chapter of LbL thin film growth measured by UV-vis and QCM.

  1. 3.3. Flame Test - must be improved.

We appreciate the reviewer’s important comment. The authors stated clearly the chapter including more discussion regarding to flame testing results.

  1. In part SEM: how the energy of the accelerator beam used?

We appreciate the reviewer’s important comment. The FE-SEM was operating at 10.0 kV.

  1. Several faults: are added or missing spaces between words: see PDF file.

We appreciate the reviewer’s important comment. As mentioned above, we have carefully checked the errors and revised this manuscript.

  1. Conclusions, the author should add some qualitative data of the results.

We appreciate the reviewer’s important comment. The authors included some quantitative results and additional descriptions in Conclusions.

  1. References: there are recent references in 2020 and 2021 treating the same subject, you can use.

We appreciate the reviewer’s important comment. As reviewer’s suggestion, the authors added some recent papers as references of this LbL FR composite study.

Round 2

Reviewer 2 Report

The manuscript can published. The authors have answered the questions.